# Hepatocytes trap and silence coxsackieviruses, protecting against systemic disease in mice

Taishi Kimura [1][✉], Claudia T. Flynn[1] & J. Lindsay Whitton [1][✉]

Previous research suggests that hepatocytes catabolize chemical toxins but do not remove microbial agents, which are filtered out by other liver cells (Kupffer cells and endothelial cells). Here we show that, contrary to current understanding, hepatocytes trap and rapidly silence type B coxsackieviruses (CVBs). In genetically wildtype mice, this activity causes hepatocyte damage, which is alleviated in mice carrying a hepatocyte-specific deletion of the coxsackievirus-adenovirus receptor. However, in these mutant mice, there is a dramatic early rise in blood-borne virus, followed by accelerated systemic disease and increased mortality. Thus, wild type hepatocytes act similarly to a sponge for CVBs, protecting against systemic illness at the expense of their own survival. We speculate that hepatocytes may play a similar role in other viral infections as well, thereby explaining why hepatocytes have evolved their remarkable regenerative capacity. Our data also suggest that, in addition to their many other functions, hepatocytes might be considered an integral part of the innate immune system.

[1] Department of Immunology and Microbiology, The Scripps Research Institute, 10550N. Torrey Pines Rd., La Jolla, CA 92037, USA.
[✉]email: tkimura@scripps.edu; lwhitton@scripps.edu

The liver captures and eradicates a variety of bacteria and viruses, a function that is currently attributed to cells residing in the hepatic sinusoids: Kupffer cells or, in some cases, sinusoidal endothelial cells[1–6]. To date, hepatocytes—by far the organ's most abundant cell type—have not been considered important in removing microbes, but herein we demonstrate that hepatocytes trap, and rapidly silence, coxsackievirus B3 (CVB3), protecting the host from systemic disease. CVB3 uses the coxsackievirus-adenovirus receptor (CAR) for binding and entry/ uncoating[7,8]. CAR is a component of intercellular tight junctions and is expressed in almost all tissues, including the liver[9]. CVB3 replicates efficiently in several tissues, such as the heart and the pancreas, leading to profound local inflammation and tissue destruction[10–13]. The liver appears to be an exception; hepatitis is rarely observed after CVB infection and, when it does occur, it appears to be age-dependent, occurring mostly in neonates[14–16]. However, paradoxically, we have reported that the livers of genetically intact CVB3-infected mice are enlarged and pale, and contain apoptotic nuclei[17], and others have recently found that primary human hepatocytes can support CVB3 replication in vitro, albeit to only a limited extent[18]. The present study was undertaken to better understand the role of the liver, and of its predominant cell type, hepatocytes, over the course of CVB3 infection. We have generated mice carrying a hepatocyte-specific deletion of the coxsackievirus-adenovirus receptor (CAR[HEP]KO mice), and have used them to demonstrate that WT hepatocytes act as a virus trap, thereby regulating CVB3-related mortality, morbidity, and systemic pathology, but at the cost of their own wellbeing. For reasons described below, we suggest that hepatocytes may internalize and silence a large variety of different viruses, diminishing pathogen burden and protecting against disease, and that this activity might be considered an integral component of the innate antiviral immune response.

## Results

### Excision of the floxed segment in CAR[HEP]KO mice is liver-specific and affects CAR protein expression only in hepatocytes.
To determine the effect of hepatocyte-specific deletion of CAR in vivo, we first generated CAR[HEP] knock out (KO) mice, by cross-breeding two extant mouse strains, Alb[Cre] mice and CAR[f/f] mice, which are described in "Materials and Methods". PCR analyses of the CAR[HEP]KO mice indicate that the deletion of the floxed CAR exon 2 was highly effective in liver, and was undetectable in the tail, pancreas, or heart (Fig. 1a, b). CAR protein was expressed at high levels in the hearts of both wildtype (WT) and CAR[HEP]KO mice, and was abundant in the WT liver, but was barely detectable in the CAR[HEP]KO liver (Fig. 1c). Alb[Cre] mice have been widely used to generate hepatocyte-specific deletion of floxed DNA segments and, by separating hepatocytes from other liver cell types, it has been shown that deletion does not take place the latter cell population[19]. Nevertheless, given the established importance of Kupffer cells in clearing certain microbes, we considered it important to confirm that these cells maintained their normal level of CAR expression in the CAR[HEP]KO mice. Flow cytometric assessment of isolated liver cells from the two mouse strains demonstrated that, in the CAR[HEP]KO mice, CAR expression was deleted from hepatocytes (Fig. 1d) but was maintained in Kupffer cells (Fig. 1e). Immunohistochemical analyses of livers from WT and CAR[HEP]KO mice was consistent with these findings, revealing that the residual CAR protein expression in CAR[HEP]KO livers was sinusoidal in location (Fig. 1f).

### Ablation of CAR from hepatocytes has no statistically significant impact on hepatic CVB3 titers.
Next, we evaluated whether hepatocytes support CVB3 replication. We began in

tissue culture, using three different hepatocyte cell lines: the SV40-transformed line H2.35; primary hepatocytes isolated from B6 mice in our laboratory; and primary hepatocytes purchased from a commercial source. CVB3 replicated to very high titers in the transformed hepatocytes, which are capable of rapid multiplication, but to much lower levels in the primary cell isolates, which do not proliferate (Fig. 2a). The latter finding mirrors the recent report in which CVB3 was shown to replicate poorly in primary human hepatocytes[18]. Next, we used a CRISPR/Cas9 approach to ablate CAR expression in H2.35 cells. Three different plasmids were used to transfect the cells: empty vector, and two plasmids (termed #2 and #3) each encoding a different sgCAR RNA. Bulk cell lines were obtained, and CAR expression was assessed (Fig. 2b); the protein was abundant in the vector control line, diminished in the sgCAR#2 line, and essentially undetectable in the sgCAR#3 cells. All three cell lines then were infected with CVB3 (moi =1), and measurements of infectious virus yield demonstrated that CAR expression is required for CVB3 infection of hepatocytes (Fig. 2c). We then evaluated viral replication in vivo, beginning by comparing CAR WT mice to CAR[HEP]KO mice. Both strains of mice were challenged with $10^4$ pfu i.p. of CVB3, and genomic RNA levels (Fig. 2d) and virus titers (Fig. 2e) in the livers were determined 1, 2, 4, and 8 days later. At each of the four-time points, genomic RNA levels were almost identical in both mouse lines, indicating that hepatocytes are not a major site of CVB3 replication in vivo. Virus titers were somewhat more variable, but at no time point was there a statistically significant difference in titer between WT and CAR[HEP]KO livers. In both strains, hepatic virus titers rose very substantially between days 1 and 4 p.i., indicating that CVB3 does replicate in the liver, albeit not in hepatocytes. The site(s) of hepatic CVB3 replication will be addressed below. A recent in vitro study showed that the constitutive expression of interferon regulatory factor 1 (IRF1) in hepatocytes prevents the replication of multiple RNA viruses, including another picornavirus, hepatitis A virus[20]. We considered it possible that IRF1 might contribute to silencing CVB3 in WT hepatocytes in vivo. Therefore, we infected WT and IRF1KO mice with CVB3 ($10^4$ pfu/mouse, i.p.) and found that, compared to WT animals, significantly higher titers of CVB3 were observed in the livers of IRF1KO mice (Fig. 2f). Similar data were observed at d1 p.i. in the pancreas and small intestine, while a lesser effect was found in the heart Supplementary Fig. 1. We also used confocal microscopy of vibratome sections to identify the viral VP1 protein in the livers of WT and IRF1KO mice at d1 p.i. VP1 signal was very rarely detectable within hepatocytes of WT mice, and instead was observed in the sinusoids (Fig. 2g and Supplementary Movie 1), suggesting that, in genetically intact animals, intrahepatic CVB3 replication may occur in, and be limited to, sinusoidal cells. In contrast, the livers of IRF1KO mice contained some hepatocytes with high levels of cytosolic VP1 (Fig. 2h and Supplementary Movie 2), indicating that CVB3, like other RNA viruses, is susceptible to the antiviral impact of this protein. Our suggestion that CVB3 replicates in hepatic sinusoids, rather than in hepatocytes, is supported by analyses of hepatic CVB3 distribution at d2 p.i. As shown in Supplementary Fig. 2, in both WT and CAR[HEP]KO mice, VP1 was present mainly in sinusoids, and the same was true for dsRNA, identified using J2 antibody.

### CAR deletion from hepatocytes leads to increased mortality, morbidity, and early viremia.
Although the hepatic virus titers were comparable in CAR WT and CAR[HEP]KO mice over the course of infection, we repeatedly and consistently observed that the mutant mice showed higher mortality ($p < 0.05$, Fig. 3a) and, as the infection proceeded, the surviving CAR[HEP]KO mice also showed increased

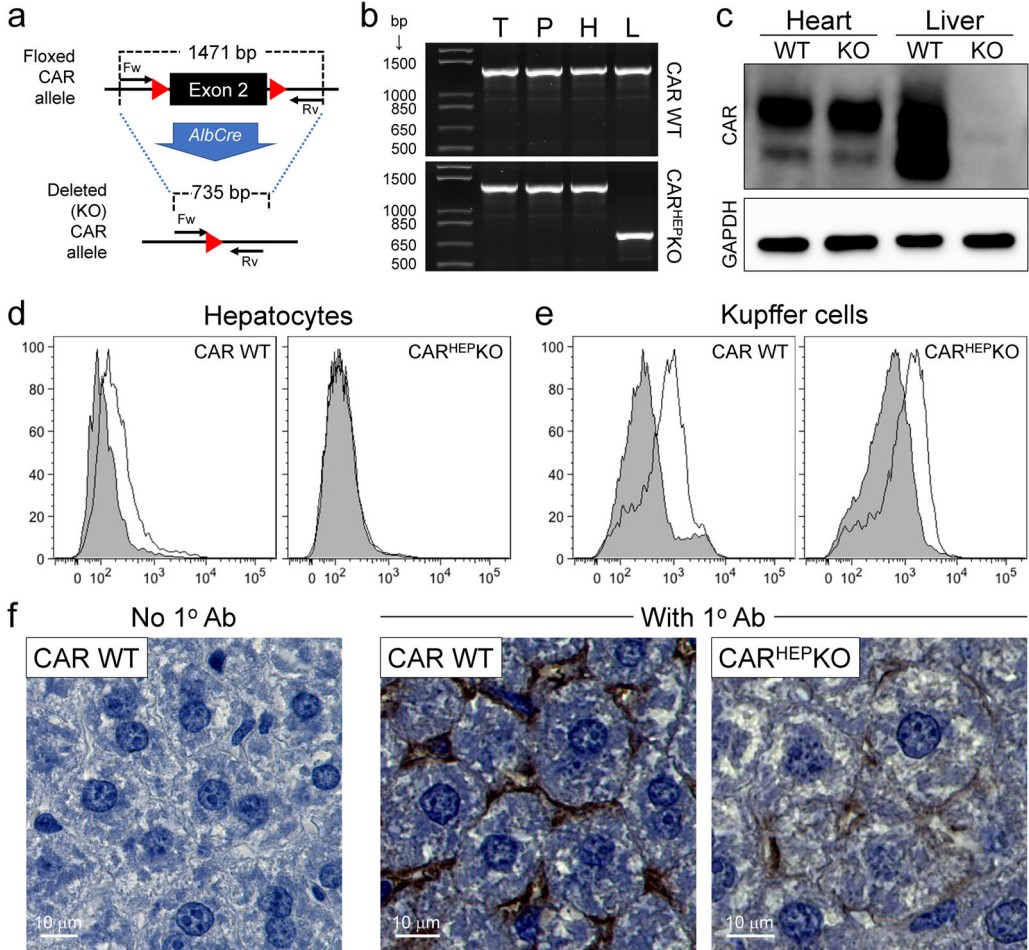

**Fig. 1 CAR^HEP^KO mice: excision of the floxed segment is liver-specific, and cell-surface CAR protein expression is ablated from hepatocytes but is maintained in Kupffer cells. a** PCR strategy for comparing the genotypes of CAR WT and CAR^HEP^KO mice. The forward (Fw) and the reverse (Rv) primers are described in "Materials and Methods". **b** the results of PCR analyses of tail (T), pancreas (P), heart (H), and liver (L) DNA are shown. The band representing deletion of CAR exon 2 is present only in the liver of CAR^HEP^KO mice. **c** hearts and livers were taken from WT mice and CAR^HEP^KO mice, and CAR protein expression was determined by western blot. Uncropped blots are shown in Supplementary Data Fig. 4. **d**, **e** CAR expression on two liver cell populations in CAR WT and in CAR^HEP^KO mice was assessed using flow cytometry. The gating strategies used are shown in Supplementary Data Fig. 5. **d** Hepatocytes (polyploid cells) and (**e**) Kupffer cells (F4/80+ cells). Isotype controls are shown in gray; CAR expression is shown as uncolored. **f** immunohistochemistry showing CAR expression in the livers of WT mice (center panel) and CAR^HEP^KO mice (right panel). Left panel = control, lacking primary (anti-CAR) antibody. Scale bars are 10 μm.

body weight loss (Fig. 3b). One of the major, and most serious, diseases associated with CVB infections is myocarditis, so we next determined if the absence of CAR from hepatocytes influenced this disorder. At d7–8 p.i., WT mice showed little or no myocardial inflammation, while CAR^HEP^KO mice showed readily detectable myocarditis (Fig. 3c, d). To determine what might underlie the above increases in systemic disease in the CAR^HEP^KO mice, we quantified the amounts of infectious virus in the blood. Strikingly, at d1 and d2 p.i., viremia in the CAR^HEP^KO mice was ~20-fold to >100-fold higher than in CAR WT mice (Fig. 3e). We reasoned that the increased early viremia in CAR^HEP^KO mice might affect virus titers in the heart, and that this might contribute to the accelerated myocarditis. Comparing CAR^HEP^KO mice to WT mice, the cardiac virus titers were similar at d1 p.i., but were >2 logs higher in CAR^HEP^KO mice at d2 and d4 p.i., and ~1 log higher at d8 (Fig. 3f), the time at which myocarditis was detectable in the mutant animals. These data suggest that the inability of hepatocytes to "absorb" CVB3 led to elevated serum titers which, in turn, resulted in exacerbation of systemic disease. A recent study proposed that hepatocytes are major producers of type 1 interferons (T1IFNs) during CVB3 infection[18], so we measured the serum IFN-α2/α4

levels of CAR^HEP^KO mice during CVB3 infection (Fig. 3g). Instead of being reduced in the CAR^HEP^KO mice, the serum IFN-α2/α4 level in these mice was markedly higher than in WT animals. We also carried out cytokine/chemokine arrays using the d2 p.i. sera from infected WT and CAR^HEP^KO mice and found that, while the difference in the production of most cytokines between CAR WT and CAR^HEP^KO mice was minimal, a few were higher in the mutant animals (Fig. 3h). Taken together, the data in Fig. 2, 3 demonstrate that, although CVB3 does not replicate efficiently in WT hepatocytes, CAR-dependent CVB3 uptake by these cells plays a key role in reducing the circulatory burden of virus, and in the consequent induction of selected cytokines and chemokines, thereby limiting mortality and morbidity, including viral myocarditis.

**CVB3 infection causes widespread CAR-dependent changes in hepatocytes, and hepatic pathology, but there is minimal inflammation.** Next, we investigated whether CVB3 uptake was harmful to WT hepatocytes in vivo. WT and CAR^HEP^KO mice were infected with CVB3 (10^4 pfu). Visual inspection of the livers of WT and CAR^HEP^KO mice at d4 p.i. revealed an obvious

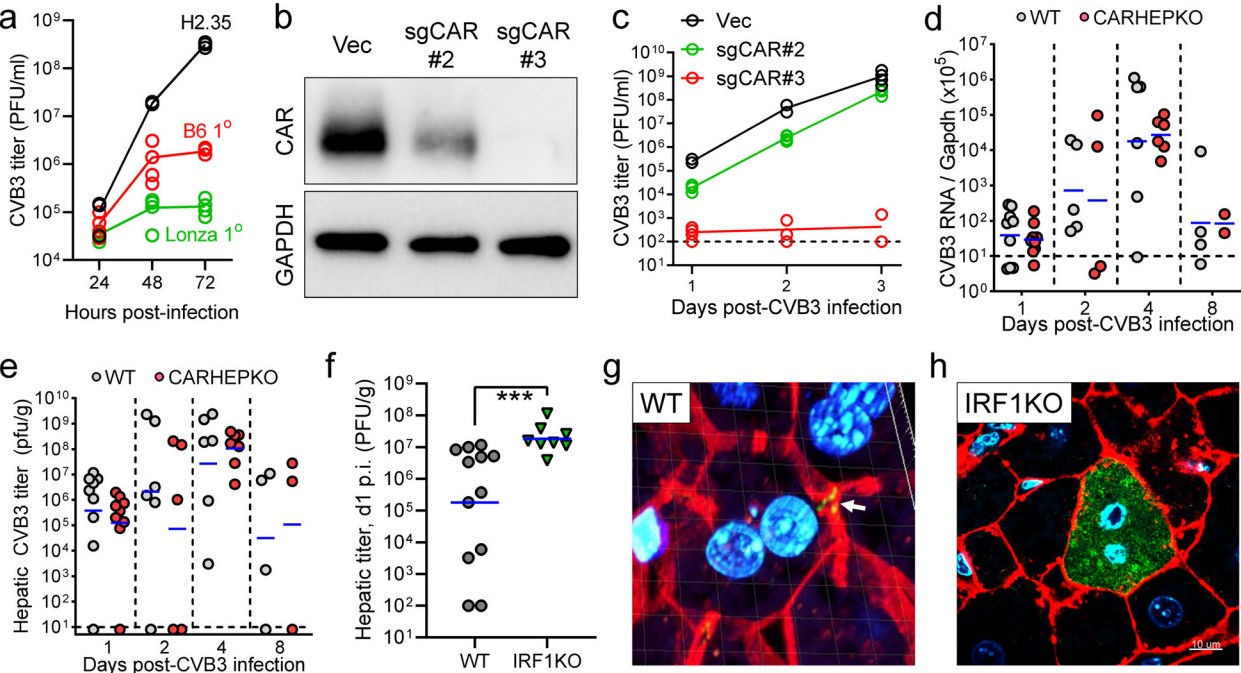

**Fig. 2 Ablation of CAR from hepatocytes has no statistically significant impact on hepatic CVB3 titers. a** H2.35 cells, and primary hepatocytes isolated in our laboratory (B6 1°) or obtained commercially (Lanzo 1°) were infected with CVB3 (moi = 1), and infectious titers in supernates were determined at the indicated time points p.i. Individual measurements are shown at all time points, and the lines show the mean values at each time point. **b** H2.35 cells were subjected to CRISPR/Cas9 modification using empty vector (Vec) or two different CAR-targeted sgRNAs (#2 & #3). Bulk cell lines were developed, and CAR protein expression was determined by western blot. Uncropped blots are shown in Supplementary Data Fig. 4. **c** The aforementioned bulk cell lines were infected with CVB3 (moi = 1) and virus yield in the supernates was determined at the indicated time points. Individual measurements are shown at all time points, and the lines show the mean values at each time point. **d, e** WT mice and CAR$^{HEP}$KO mice were challenged with CVB3 ($10^4$ pfu, i.p.) and livers were harvested at the indicated time points. Blue lines indicate geometric mean values. **d** Viral genome numbers were determined by qPCR, and **e** infectious titers were measured by plaque assay. Blue horizontal lines = geometric means. For both panels, at each time point there was no statistically significant difference between the WT and CAR$^{HEP}$KO values. **f–h** WT and IRF1KO mice were infected with CVB3 ($10^4$ pfu, i.p.) and 24 h later the mice were sacrificed, and **f** hepatic CVB3 titers were determined; blue lines indicate geometric mean values. Vibratome sections of livers were cut, and probed for CVB3 VP1 expression using an antibody. **g** VP1 (green) was not found in WT hepatocytes, but signal (white arrow) was observed within sinusoids (blue = nuclei; red = F-actin). This image has a fine white grid of squares, each side of which is 5 μm (as shown in Supplementary Movie 1h, Occasional IRF1KO hepatocytes had abundant cytosolic VP1. Scale bar is 10 μm.

difference; the former were pallid, while the latter appeared relatively normal (Fig. 4a). The macroscopic change in the WT livers was accompanied by elevated serum ALT levels at d4 p.i. (Fig. 4b), indicative of hepatocyte death[21]. In contrast, serum ALT levels did not markedly increase during CVB3 infection of CAR$^{HEP}$KO mice, indicating that uptake of the virus by hepatocytes is required for cell damage. At 4 days p.i., when ALT levels were increased, a large proportion of WT hepatocytes are actively responding to CVB3 infection; oil red O staining of vibratome sections identified steatosis, indicative of alterations in lipid metabolism (Fig. 4c), and the F-actin cytoskeleton of many WT hepatocytes was disordered (Fig. 4d). These changes were largely mitigated when the hepatocytes lacked CAR. Despite the damage to WT hepatocytes, histological analyses showed minimal inflammation in WT mice (Fig. 4e); this observation confirms others' findings that, in genetically intact animals, CVB3 does not trigger florid hepatitis[18,22]. We also performed two assays to assess hepatic apoptosis: TUNEL, and immunohistochemistry to detect cleaved caspase 3 (Supplementary Fig. 3). The TUNEL assay showed that apoptotic cells were present, albeit at low numbers, in liver sections from CVB3-infected WT mice, but they were almost undetectable in infected CAR$^{HEP}$KO livers. The cleaved caspase 3 data confirm the TUNEL findings; CVB3 rarely triggers apoptosis in WT livers. Moreover, the data support and extend the conclusion drawn from Fig. 4e; at 4 days p.i., when ALT levels are highest, hepatic inflammation is

minimal, and this is true even adjacent to the infrequent apoptotic foci.

**During CVB infection, pDCs are key sources of systemic IFN-α production, and confer significant, but incomplete, protection.** Finally, since CVB uptake by hepatocytes is not required for the observed pulse of T1IFNs (Fig. 3g), we sought to determine the cellular origin of these serum cytokines. Plasmacytoid dendritic cells (pDCs) are the obvious candidate[23], so we examined the impact of pDCs on CVB3 infection using CLEC4C-DTR mice, in which diphtheria toxin (DT) injection depletes pDCs of the animals[24]. DT injection effectively depleted both the splenic and the hepatic pDC populations (Fig. 5a). Following challenge with $10^4$ pfu i.p. of CVB3, pDC-intact controls and pDC-depleted animals showed comparable fecal virus titers with similar rates of virus clearance (Fig. 5b). Serum IFN-α production was significantly decreased in the pDC-depleted mice at d2 and d4 p.i. (Fig. 5c) indicating that, in genetically intact mice, the majority of systemic IFN-α production that is induced by CVB3 infection is attributable to the presence of pDCs. pDC-depleted mice lost almost twice as much body weight compared to the pDC-intact animals during the early phase (d2 to d4) after CVB3 infection (Fig. 5d), suggesting that pDCs may contribute to host defense against CVB3 infection at these early time points. We also assessed the protective role of pDCs against a lower CVB3

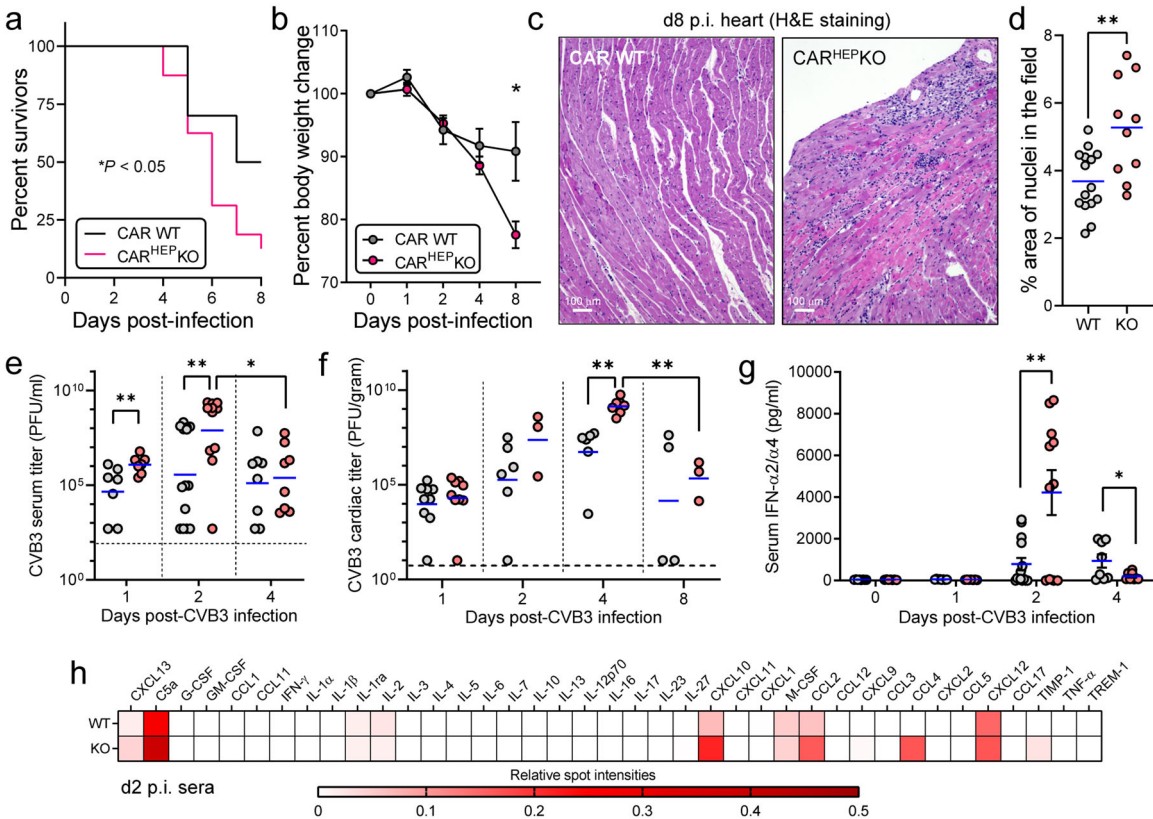

**Fig. 3 CAR deletion from hepatocytes leads to increased mortality, morbidity, and early viremia. a** Survival curve of CAR WT ($n = 10$) and CAR^HEP^KO ($n = 16$) mice after CVB3 infection ($10^4$ pfu). **b** Body weight changes of the CAR WT ($n = 10$) and CAR^HEP^KO ($n = 16$) mice after CVB3 infection ($10^4$ pfu) (Mean ± SEM). **c** Hematoxylin/eosin-stained sections of hearts from WT and CAR^HEP^KO mice, harvested at 8 day after CVB3 infection ($10^4$ pfu). Scale bars are 100 μm. **d** The extent of myocarditis in hearts obtained at days 7 or 8 p.i. was quantified by using ImageJ to identify the percentage of tissue area represented by nuclei. WT hearts, $n = 3$; CAR^HEP^KO hearts, $n = 2$; and, from each heart, five 20x fields were evaluated. Blue horizontal lines indicate mean values. **e, f, g** Gray circles = CAR WT; pink circles = CAR^HEP^KO. **e** Viremia and (**f**) cardiac virus titers were determined by plaque assays at the indicated days after CVB3 infection ($10^4$ pfu). Blue horizontal lines = geometric means. **g** Serum IFN-α2/α4 levels of WT ($n = 8$-14) and CAR^HEP^KO ($n = 7$-11) at the indicated days after CVB3 infection ($10^4$ pfu) determined by ELISA (mean ± SEM). **h** Cytokine arrays using the sera from CAR WT and CAR^HEP^KO mice. Each value was normalized by the reference spot and the relative intensities are shown as heat map. $P$ values less than 0.05 were considered significant and are indicated in panels as follows: * $0.05 > p > 0.01$; ** $0.01 \geq p > 0.001$.

challenge (500 pfu). At this dose, all pDC-intact mice survived the infection, but the pDC-depleted mice started showing mortality from d4 p.i. ($p = 0.0299$, Fig. 5e). pDC-intact mice efficiently eradicated the fecal CVB3 as early as d8 p.i., while the pDC-depleted mice failed to do so (Fig. 5f). As was observed following the $10^4$ pfu infection experiments, more serious body weight loss was observed in the pDC-depleted mice at 2–4 days after 500 pfu of CVB3 infection (Fig. 5g). Collectively, these data indicate that pDCs are a key cellular source of systemic T1IFNs during CVB3 infection, and the cells contribute to protection against both morbidity and mortality.

## Discussion

We show here that mice carrying a hepatocyte-specific deletion of the coxsackievirus-adenovirus receptor (CAR^HEP^KO mice) are unable to capture circulating CVB3, leading to a dramatic early rise in blood-borne virus, followed by accelerated systemic disease and increased mortality. In genetically intact mice, this trapping and rapid silencing of virus lead to widespread alterations in hepatocyte morphology, and to short-lived hepatic pathology. Thus, hepatocytes act as a trap for CVBs, thereby reducing systemic viral titers and disease, but they do so to their own detriment. To the best of our knowledge, this is the first in vivo demonstration that hepatocytes can act as a sponge, trapping and suppressing viruses without triggering overt hepatitis.

CVB3 productively infects cardiomyocytes and pancreatic acinar cells, causing profound myocarditis and pancreatitis. Others have described mice carrying specific CAR deletions from those cell types[12,13], and the in vivo effects of those deletions differ greatly from those we report herein for hepatocytes. Specific deletion of CAR from cardiomyocytes or pancreatic acinar cells dramatically decreases CVB3 titers in the related tissues (heart and pancreas, respectively), and protects the infected animals from the relevant inflammatory disease (i.e., from myocarditis and pancreatitis, respectively). However, the deletions have no known statistically significant effect on either viremia or systemic disease (i.e., disease in other CAR-intact tissues in the animal). In contrast, we show here that hepatocyte-specific CAR ablation does not markedly affect viral titers in the liver (Fig. 2d, e) and that it does not prevent local inflammation (because there is no detectable hepatitis, even in the WT livers—Fig. 4e). We also show that hepatocyte-specific CAR ablation results in an early, and massive, increase in viremia (Fig. 3e) and enhances systemic disease in distant organs (myocarditis), even though the hearts of our mice are CAR-intact (Fig. 3c). Thus, the two effects of CAR deletion that we report—CVB3 uptake/silencing and protection against systemic disease—are specific to hepatocytes.

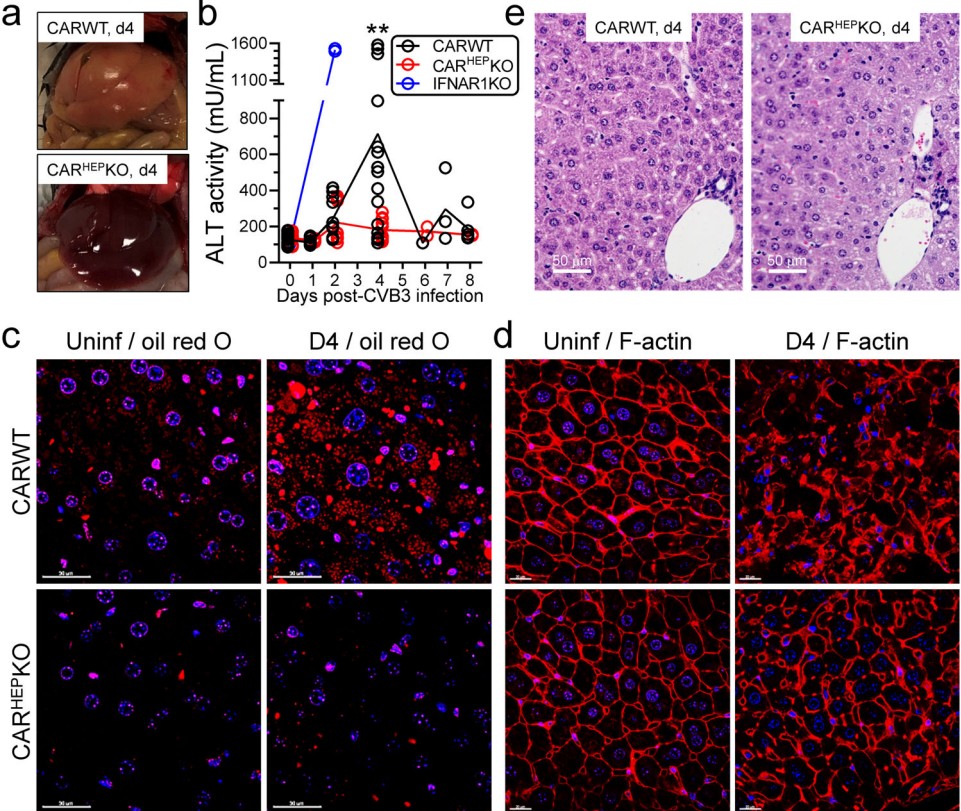

**Fig. 4 CVB3 infection causes widespread CAR-dependent changes in hepatocytes, and hepatic pathology, but there is minimal inflammation. a** Macroscopic images of the livers of representative mice, taken at d4 after CVB3 infection ($10^4$ pfu). **b** ALT activity in sera from the indicated mice after CVB3 infection ($2 \times 10^4$ pfu for IFNAR1KO mice and $10^4$ pfu for the others; IFNAR1KO mice were used as a control for the assay). WT ($n = 1$–22), CAR$^{HEP}$KO ($n = 2$–15), and IFNAR1KO ($n = 2$) (Mean ± SEM). ** $0.01 \geq p > 0.001$. Vibratome sections of WT or CAR$^{HEP}$KO livers were stained with (**c**) oil red O or (**d**) phalloidin (red), which binds to beta-actin. White scale bars represent 30 μm (**c**) and 20 μm (**d**). **e** Histological sections of representative livers from WT and CAR$^{HEP}$KO mice at 4 days after CVB3 infection ($10^4$ pfu), stained with Hematoxylin/Eosin. Scale bars are 50 μm.

Almost all somatic cells mount innate immune responses to viruses whose productive replication they support. Hepatocytes are no exception, and others have shown that, as expected, they respond to infection by hepatitis viruses, and their pro-inflammatory responses lead to extensive local inflammation (i.e., florid hepatitis). However, the hepatocyte activity that we describe here is very different from this standard innate cellular response, because it applies to a virus, CVB3, that is not considered to be hepatotropic, does not productively infect the cells, and does not cause hepatitis. We show that IRF1 appears important to the silencing of CVB3 in hepatocytes (Fig. 2f–h), but the hepatocyte gene product(s) that actively suppresses viral replication will, quite probably, vary depending on the virus that is involved. By acting as a charnel house for coxsackieviruses, hepatocytes play a vital part in protecting extrahepatic tissues, and the host animal, from the ravages wrought by these pathogens. However, the WT hepatocytes, despite not being productively infected, are damaged by their ingestion of CVB3: the liver is macroscopically abnormal, a large number of hepatocytes are steatotic, and ALT is elevated (Fig. 4). These effects are mitigated by the absence of CAR from the hepatocyte cell membrane, demonstrating that these pathological effects are initiated by viral uptake. Thus, following i.p. infection, hepatocytes act as a shield against CVB3, but they also may do so during natural infections by many other non-hepatitis viruses. It is well recognized that systemic infections by a variety of viruses (e.g., herpesviruses, parvoviruses, influenza viruses) can be accompanied by transient elevations of serum transaminases[25], and our observations offer a simple explanation; perhaps these viruses, too, are absorbed and silenced by hepatocytes, to the cells' own detriment. In that light, COVID-19, caused by a novel coronavirus (nCov-19, aka SARS-CoV-2), is primarily a respiratory disease, but up to ~50% of patients—in particular those with the most serious signs and symptoms—have elevated transaminases[26]. Examination of the liver of a fatal case of COVID-19 revealed steatosis, but no clear evidence of SARS-CoV-2 infection[27]. Moreover, raised transaminase levels were observed in patients infected by the original SARS coronavirus (SARS-CoV), and also in patients infected with the Middle East respiratory syndrome (MERS) coronavirus and, in both cases, viral particles were undetectable in the liver[28,29]. These observations are consistent with our suggestion that, during the early days of infection, hepatocytes protect the host by trapping and silencing a broad spectrum of viruses, sacrificing themselves for the good of the organism. We propose that this previously unidentified hepatocyte activity might be considered an integral part of the innate response to viral infection. We also note that hepatocytes have a unique capacity for regeneration. After surgical removal of two-thirds of the organ, a mouse liver regains its starting weight within ~7 days[30], and it has been estimated that a single adult mouse hepatocyte could undergo sufficient doublings to generate 50 mouse livers[31]. Thus, we speculate that this remarkable replicative capacity of hepatocytes may be related, at least in part, to the function described herein. Throughout evolution, animals have been faced with constant viral onslaught and may have selected for one cell type—the hepatocyte—to develop two inter-related functions. These cells

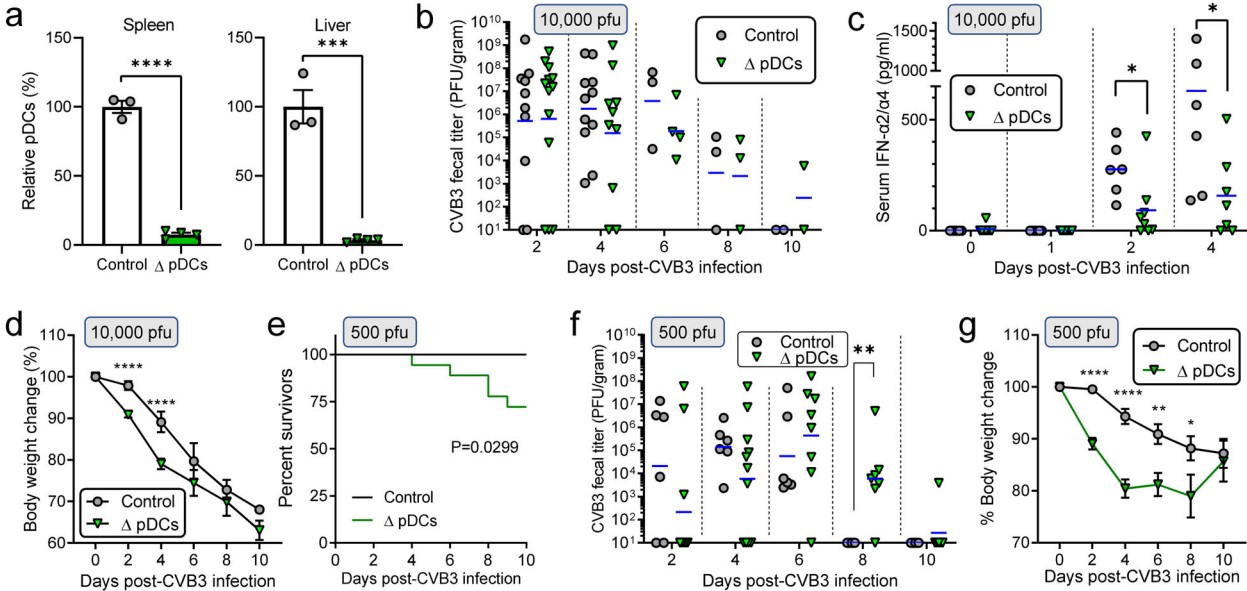

**Fig. 5 During CVB infection, pDCs are key sources of systemic IFN-α production, and confer significant, but incomplete, protection. a** Relative pDC populations in spleens and livers of control ($n = 3$) and pDC-depleted (Δ pDCs) ($n = 4$) CLEC-DTR mice after two injections of DT (2 µg) (mean ± SEM). The value of control mice was set as 100%. The gating strategies used are shown in Supplementary Data Fig. 5. **b** Fecal virus titers of control ($n = 11$) and pDC-depleted (Δ pDCs) ($n = 13$) mice at the indicated days after CVB3 infection ($10^4$ pfu) determined by plaque assays. Blue horizontal lines, in this and subsequent panels, = geometric means. **c** Serum IFN-α2/α4 levels determined by ELISA (Mean ± SEM). WT ($n = 6$) and CAR$^{HEP}$KO ($n = 8$). **d** Body weight changes of control ($n = 12$) and pDC-depleted (Δ pDCs) ($n = 15$) mice after CVB3 infection ($10^4$ pfu) (mean ± SEM). **e** Survival curve of control ($n = 15$) and pDC-depleted (Δ pDCs) ($n = 18$) mice after CVB3 infection (500 pfu). **f** Fecal virus titers of control ($n = 6$) and pDC-depleted (Δ pDCs) ($n = 11$) at the indicated days after CVB3 infection (500 pfu) determined by plaque assays. **g** Body weight changes of the control ($n = 15$) and pDC-depleted (Δ pDCs) ($n = 11$) mice after CVB3 infection (500 pfu) (mean ± SEM). $P$ values less than 0.05 were considered significant, and are indicated in the panels as follows: * $0.05 > p > 0.01$; ** $0.01 \geq p > 0.001$; *** $0.001 \geq p > 0.0001$; **** $p \leq 0.0001$.

can absorb many different blood-borne viruses, and quickly silence them, but being damaged in the process. Concurrently, their uninfected neighbors can rapidly divide, maintaining the organ's other biological functions, and thereby further limiting host disease.

The protective activity that we describe is dependent upon the cells' expressing the CVB3 receptor, CAR. Others have previously designed CAR-based traps to ameliorate CVB3-induced diseases. Injected, or ectopically expressed, soluble IgG-combined forms of recombinant human CAR strongly inhibit CVB3 replication and protect infected mice from cardiac inflammation[32,33]. Furthermore, the transgenic expression of CAR on erythrocytes attenuates CVB infection and viral pathogenesis[34]. Our data indicate that these human-made CAR traps inadvertently recapitulate the analogous trap function of hepatocytes. We show here that this "natural trap" limits early viremia, and subsequent morbidity and mortality, including the devastating disease of viral myocarditis. Our observations may offer a new therapeutic avenue, based on the exploitation of this naturally designed viral trap. By artificially increasing hepatocyte uptake of pathogens, and thus their clearance, we may protect the host against a variety of systemic diseases. Although such therapy would probably be harmful to the hepatocytes that engulfed the microbes, we predict the long-term effects on the organ and on the host should be minimal.

## Methods

**Ethics statement.** All animal experiments were approved by The Scripps Research Institute (TSRI) Institutional Animal Care and Use Committee (protocol number 09-0131-3) and were carried out in accordance with the National Institutes of Health's Guide for the Care and Use of Laboratory Animals.

**Mice.** CAR-floxed mice[35] were a generous gift from Dr. Robert Ross at UC San Diego. CLEC4C-DTR mice[24] (JAX 014176), Albumin-Cre transgenic mice[36] (JAX 003574; herein, Alb$^{Cre}$) and IRF1KO mice[37] (JAX 002762) were purchased from

the Jackson laboratory. To generate CAR$^{HEP}$KO mice, CAR-floxed (CAR$^{f/f}$) mice were mated with Albumin-Cre transgenic mice (Alb$^{Cre}$) to generate CAR$^{f/+}$ Alb$^{Cre}$ mice. The doubly mutant mice were then intercrossed to obtain CAR$^{f/f}$ Alb$^{Cre}$ mice (i.e., CAR$^{HEP}$KO mice). Mouse genotypes were assessed as shown in Fig. 1a, using the forward primer (5'CAAAGGTACCACAACCCTTG3'), and the reverse primer (5'TGTGGTGCAGGCTGTCTTCA3'). The lines were maintained by mating CAR$^{HEP}$KO mice with CAR$^{f/f}$ mice to obtain additional CAR$^{HEP}$KO mice, as well as CAR$^{f/f}$ Cre$^−$ littermate controls, which are genetically CAR-intact. The two types of offspring were born at a ~1:1 ratio, and the mice were phenotypically indistinguishable, indicating that deletion of CAR from hepatocytes is well-tolerated; this contrasts with the effects of total CAR deletion, or deletion specifically from cardiomyocytes, both of which are embryonically lethal[35,38,39]. For diphtheria toxin (DT)-mediated pDC depletion models, CLEC4C-DTR(−) and CLEC4C-DTR(+) transgenic mice were injected with PBS or 2 µg of DT (Millipore SIGMA #D0564) twice (at days −1 and +1 of CVB3 infection) i.p. The DT-treated CLEC4C-DTR (+) mice were used as the pDC-depleted mice and the DT-treated CLEC4C-DTR (-) mice and the PBS-treated mice were used as the control pDC-intact mice for the pDC-depleted mice. C57BL/6 mice were purchased from the TSRI rodent breeding colony and the Jackson laboratory (JAX 000664) and used as the control wild-type for IRF1KO mice. For in vivo virus infection, the animals were bred and housed in Biosafety Level-2 (BSL-2) and only male animals were used for these studies. Each experimental group was generated by matching their ages (8–14-week old) between the groups in each experiment.

**Cells.** HeLa cells for plaque assays were maintained with DMEM (Life Technologies #10313-021) supplemented with 10% FBS (Omega Scientific #FB-02) and 1x Penicillin-Streptomycin-Glutamine (Thermo Fisher Scientific #10378016) in 5% $CO_2$ incubator at 37 °C. H2.35 cells were purchased from ATCC (#CRL-1995), and were maintained in a 10% $CO2$ incubator at 33 °C in the following medium: DMEM, low glucose, GlutaMAX Supplement, pyruvate (Life Technologies #10567-014), supplemented with 200 nM dexamethasone (SIGMA Aldrich, #D4902), 4% FBS and 1× Penicillin-Streptomycin (Thermo Fisher Scientific #15140148). Cryopreserved C57BL/6 Mouse Hepatocytes were purchased from Lonza (#MBCP01) and were maintained in the maintain media provided by the manufacturer in a 5% $CO2$ incubator at 37 °C. For isolation of primary hepatocytes, we modified a published procedure[40], dispensing with the use of peristaltic pump perfusion. In brief, mice were sacrificed and perfused 3× with (1) Dulbecco PBS, (2) warmed Buffer 1 (142 mM NaCl, 6.7 mM KCl, 10 mM HEPES, 0.226 mM BSA, at pH 7.4) and (3) warmed Collagenase D (Roche, Cat# 11088866001) in Buffer 2 (66.74 mM NaCl, 6.71 mM KCl, 6.31 mM CaCl2, 100 mM HEPES, 0.226 mM BSA, 0.03 mM

phenol red, at pH 7.4). Liver was then perfused directly via portal vein with Collagenase D in Buffer 2 allowing the solution to fill all lobes of the liver. The liver was isolated, the gall bladder was removed, and the liver was placed on a small petri dish. The liver was gently shaken, allowing the perfused fluid, plus cells, to be collected in the dish. The liver was then cut into small pieces, and the contents of the petri dish were transferred to a 25 ml Erlenmeyer flask. Oxygen (Praxair, Cat# UN1072) was added, sealed with a rubber top and parafilm and incubated on shaker at 150 rpm at 37 °C for 30 min. After digestion, cell suspension was transferred through a 70 μm strainer into a 50 ml conical tube and washed twice with warmed Buffer 3 (137 mM NaCl, 4.7 mM KCl, 0.66 mM MgSO₄, 1.62 mM CaCl2, 10 mM HEPES, 0.266 mM BSA, at pH 7.4). Cells were spun at 800 rpm for 3 min after each wash, then were resuspended in hepatocyte culture media: Williams E medium, no phenol red (ThermoFisher Scientific, Cat # A1217601) supplemented with hepatocyte maintenance pack (ThermoFisher Scientific, Cat # CM4000). Cells were counted and plated onto 24-well collagen I coated plates (Corning, Cat# 354408) and maintained in a 5% CO2 incubator at 37 °C. Culture media was replaced at 24 h and 48 h post culture. After 5 days of culture, cells were used as primary hepatocytes for further applications.

**Viruses.** The wild-type CVB3 used in these studies is a plaque purified isolate (designated H3) of the myocarditic Woodruff variant of CVB3. Plasmid pH 3, encoding a full-length infectious clone of this virus, was provided by Dr. Kirk Knowlton (University of California, San Diego). Mice were infected intraperitoneally (i.p) with the indicated dose of CVB3 and their survival and body weight were monitored during the course of infection. Mice were bled on the indicated days p.i. and, at the end of experiment, were sacrificed, and their blood and tissues were removed for the subsequent assays.

**Mouse serum samples.** For serum isolation, whole blood was collected into MiniCollect serum separation gel tubes (Greiner Bio-One, #450472) and centrifuged for 10 min at 3000g room temperature. The sera were stored at −20 °C until use.

**Plaque assays.** Feces were collected at the indicated times p.i. At the time of sacrifice, mice were perfused with DPBS and then pancreata, livers, hearts, and small intestines were isolated, weighed, and homogenized in 1 ml Dulbecco Modified Eagle Medium (DMEM). Hepatocyte supernatants were collected at indicated time points. CVB3 plaque assays were performed as described previously[41] with slight modification. Briefly, $1.7 \times 10^5$ HeLa cells were seeded 24 h prior to inoculation in a 24-well plate. 100 μl of serial diluted inoculum was added to the cell monolayers. Plates were shaken 10–15 min interval for 1 h to prevent dryness of monolayer. One hour later, inoculum was removed and cells were overlaid with DMEM containing 0.6% agar (Fisher Bioreagent #BP1423-500). 42 h post-inoculation, cells were fixed with 25% Acetic Acid (Fisher Chemical #A38SI-212)/75% Methanol (Fisher Bioreagent #BP1105-4) and stained with 0.25% Crystal Violet (SIGMA Aldrich #C0775). After washing Crystal Violet solution, plaques were counted and PFU/ml and PFU/gram were calculated based on dilution rates and tissue weigh, respectively.

**CRISPR/Cas9-mediated gene editing in H2.35 cells.** $2 \times 10^5$ H2.35 cells were seeded onto six-well plates. 24 h later, pX459 ver 2.0 (Addgene, #62988) encoding sgCARs, or empty (control) was transfected into H2.35 cells and incubated for further 24 h. Then culture medium was changed to the media containing puromycin (2 μg/ml) and Cas9-expressing cells were selected for three days. After drug selection, puromycin was removed from culture media and cells were recovered. These bulk gene-edited cells were used for in vitro studies.

**BCA assays and western blotting.** Tissues were homogenized in freshly prepared RIPA buffer (EMD Millipore #20-188) containing the Protease inhibitor cocktail (Roche/Genentech #11697498001) and the Halt Phosphatase Inhibitor Cocktail (Pierce/Thermo Fisher Scientific #1862495). After centrifugation for 10 min at 4 °C, debris was discarded and protein concentration in the supernatant was determined by Pierce BCA Protein Assay Kit (Thermo Fisher Scientific #23225). Colorimetry was measured using plate reader, Victor X3 (Perkin Elmer). Five microgram of the total cell lysates were mixed with Laemmli Sample Buffer (Bio-Rad #161-0747) and 10% of 2-Mercaptoethanol (SIGMA Aldrich #M6250), boiled at 98 °C for 5 min and used for western blotting. Transfer of the proteins to membrane was performed using Transblot Turbo RTA Transfer Kit (Bio-Rad #170-4272) as follows. After developed on SDS gels, proteins were transferred on PVDF membranes (Bio-Rad, a component of the Transfer Kit) by using Transblot Turbo system (Bio-Rad). Membranes were then blocked with 1% skim milk (EMD Millipore #115363) for an hour, and overnight with the relevant diluted primary antibodies [anti-CXADR/CAR antibody (provided by Dr. Klingel, University Hospital Tübingen, Germany), and anti-GAPDH antibody (clone 6C5), EMD Millipore #MAB374]. Then, membranes were washed three times with Tris buffer Tween 20 (TBST) and incubated with diluted secondary antibodies (HRP-conjugated donkey anti-rabbit IgG or sheep anti-mouse IgG, GE Healthcare Life Sciences #NA934 & NA931). One hour later, membranes were washed again three times with TBST, then protein–antibody complexes were visualized by Super Signal ELISA Femto

Maximum Sensitivity Substrate (Thermo Scientific #37074). Images were captured with the ChemiDoc imager (Bio-Rad).

**Real-time RT-PCR.** RNA was isolated from tissue and cell suspensions using the RNeasy Mini Kit (QIAGEN, # 74104), and 1–2 μg of RNA was reverse transcribed using iScript Reverse Transcription supermix (Bio-rad, #1708841). Real-time PCR was performed using Power SYBR Green PCR mastermix reagent (Applied biosystems, #4367659) with specific primer sets; CVB3, Forward: CACACTCCGAT CAACAGTCA and Reverse: GAACGCTTTCTCCTTCAACC. Gapdh, Forward: AGGTCGGTGTGAACGGATTTG and Reverse: TGTAGACCATGTAG TTGAGGTCA. All the values were normalized to the values of *Gapdh*.

**Immunofluorescence, immunohistochemistry, and histological analysis.** Mice were perfused with DPBS, and tissues were harvested, fixed using buffered zinc formalin at room temperature overnight. For immunofluorescence, 70 μM sections of liver were cut with a Leica VT 1000S Vibratome. For VP1, nuclei and F-Actin staining, sections were antibody-labeled using the MAX FluorTM Mouse on Mouse Fluorescent Detection kit (MaxVision Biosciences #MF01-S). Briefly, the tissue sections were washed with PBS for 5 min three times and permeabilized with 0.5% Triton/PBS for 30 min at room temperature. Then, enough Protein Blocking Solution (Reagent 1) was added and sections were incubated for 10 min at room temp. Excess Reagent 1 was removed and sections were blocked with MaxMOM Blocking reagent (Reagent 2) for 60 min at room temperature. Sections were washed with 0.3% Triton/PBS for 5 min three times. Sections were incubated with primary antibody [Mouse anti-Enterovirus VP1 (Clone 5-D8/1, Mediagnost Germany #M47) or the dsRNA-specific J2 antibody (Scicons #10010500) diluted to 1:2000 in 0.3% Triton/PBS] for one hour at room temperature and then at 4 °C overnight. Next day, sections were washed with 0.3% Triton/PBS for 5 min three times and enough Fluorescent Signal Enhancer (Reagent 3) was added and sections incubated for 30 min at room temperature. Sections were then rinsed with 0.3% Triton/PBS for 5 min three times. A working solution of concentrated Max Fluor 488 Labeled Linker (Reagent 4) diluted in Fluorescent Diluent (Reagent 5) (1:400) was added to sections and incubated in the dark for one hour at room temperature. After further washing with 0.3% Triton/PBS, sections were incubated with Alexa Fluor 568 Phalloidin (Thermo Scientific #A12380) (1:40) in PBS at 4 °C overnight to label F-actin. After incubation, sections were rinsed with PBS for 5 min three times, then counterstained with Hoechst 33342 (1:10,000) (Thermo Fisher Scientific #H3570). For oil red O staining, sections were stained using an oil red O stain kit (Abcam, #ab150678) according to the manufacturer's instruction, then counterstained with Hoechst 33342 (1:10,000). For the other histological analyses, formalin-fixed tissues were paraffin embedded and 3 μm sections were cut. For standard histological analyses, sections were stained with Hematoxylin-Eosin. For immunohistochemistry, sections were deparaffinized with Xylene for 10 min, 100% Ethanol for 10 min, 90% ethanol for 3 min, 80% ethanol for 3 min, 70% ethanol for 3 min, then washed with TBS for 5 min. The deparaffinized sections were either incubated in 10 mM sodium citrate buffer (pH 8.34) at 80 °C for 30 min (cleaved caspase-3 staining and CAR staining) or directly subjected to TUNEL assays. For cleaved caspase-3 staining and CAR staining, Cell and Tissue Staining Kit Anti-Rabbit HRP-DAB system (R&D Systems #CTS005) was used according to the manufacturer's instructions, with either anti-cleaved caspase-3 antibody (Cell Signaling Technology #9664 T) or anti-CAR/CXADR antibody (Sinobiological #50019-R001). TUNEL assays were performed using a TUNEL Assay Kit - HRP-DAB (Abcam, #ab206386) according to the manufacturer's instructions. All sections were mounted with ProLong Gold Antifade Mountant solution (Invitrogen #P36930) for imaging.

**Imaging and analysis.** Confocal images were captured using a Zeiss LSM 780 Laser Confocal Scanning Microscope running Zen 2009 Zeiss software suite. Representative regions within each vibratome section of the tissues were scanned and reconstructed for analysis. Exposure and image acquisition settings were identical for all sections. Histological images were captured with an BZ-X710 inverted microscope (KEYENCE) using BZ-X Viewer software (KEYENCE). Myocarditis in the images at x10 magnification was quantified using the particle analysis command in ImageJ software to determine the percentage area, in multiple cardiac sections, that was represented by nuclei.

**Enzyme-linked immunosorbent assay (ELISA).** Serum IFN-α2/α4 levels were measured using mouse IFN-α platinum ELISA kit (Thermo Scientific #BMS 6027) according to the manufacturer's instruction. Briefly, all tested sera were diluted in Assay Buffer (1:2, v/v). Microwell strips were washed twice, and appropriate volumes of Assay Buffer and standards/samples were added to each well. Then, Biotin-Conjugate was further added to each well, and the plate was incubated for 2 h at room temperature on a microplate shaker. After washing with Wash Buffer four times, Streptavidin-HRP was added to each well and the plate was incubated for one hour at room temperature on a microplate shaker. After washing four times, TMB Substrate Solution was added to each well and the plate was incubated for 15–30 min at room temperature in the dark. Once the colorimetric reaction in standard wells has become obvious, the reaction was stopped with Stop Solution

and the color intensity was measured at 450 nm using plate reader, Victor X3 (Perkin Elmer).

**Serum cytokine array**. Sera from CAR WT and CAR[HEP]KO mice were used for cytokine array. The array was performed using Proteome Profiler Mouse Cytokine Array Panel A Kit (R&D systems #ARY006) according to the manufacturer's instruction. Briefly, membranes were blocked for one hour at room temperature. Meanwhile, samples were incubated with detection antibody for one hour at room temperature. Then, the blocked membranes were incubated in the sample/detection antibody mixtures overnight at 4 °C. Next day, the membranes were washed three times and incubated with diluted Streptavidin-HRP for 30 min. at room temperature. After washing three times, the membranes were incubated with Chemi Reagent Mix and each spot was visualized on ChemiDoc™ touch imaging system (Bio-Rad). The signal intensities of each spot were calculated using ImageJ software, and each value was normalized against the mean values of reference spots (pre-defined by the manufacturer).

**ALT activity assay**. Serum ALT activity was measured using the alanine aminotransferase activity colorimetric/fluorometric assay kit (BioVision, #K752-100) according to the manufacturer's instruction. Briefly, serum samples were diluted at 1:1,000 in ALT Assay Buffer and mixed with 100 Reaction Mix. Then, the fluorescence was measured at Ex/Em = 531/595 nm using plate reader, Victor X3 (Perkin Elmer) at least two different time points. ALT activity was calculated by $(B \times \text{dilution factor})/[(T_2 - T_1) \times 0.02]$, where $B$ is the pyruvate amount from pyruvate Standard Curve (in nmol), $T_1$ and $T_2$ are the time of the first and the second reading (in min), respectively, and shown as mU/ml.

**Flow cytometry**. To determine the surface CAR expression on hepatocytes and Kupffer cells, mice were sacrificed and perfused with DPBS. Livers were isolated, cut into small pieces and incubated in digestion solution [Collagenase D (Roche, #11088860001) 0.625 mg/ml, DNase (Sigma-Aldrich, #10104159001) 0.1 mg/ml, Collagenase V (Sigma-Aldrich, #C9263) 0.85 mg/ml, Dispase (Sigma-Aldrich, #D4693) 1 mg/ml in RPMI] by shaking at 175 rpm at 37 °C for 25 min. Then, the digested tissues were removed by a 100 μm mesh and set aside for hepatocyte analysis. Leukocytes in the flow through were washed with RPMI twice, then red blood cells were lysed with 0.83% NH4Cl. The cells were passed through a 70 μm cell strainer, treated with anti-CD16/32 (BD Biosciences #553142) for Fc-blocking, and subjected to CAR staining (below). The digested liver tissues for hepatocyte analysis were mechanically disrupted, and the cell pellets were washed twice with RPMI, then treated with Hoechst 33342 (Thermo Fisher Scientific #H3570) 15 μg/ml for 30 min on ice. After washing with FACS buffer [2% FBS (Omega Scientific #FB-02) in PBS] twice, both the Fc-blocked hepatic lymphocytes and the hepatic cell pellets were treated with either control purified mouse IgG1 (BioLegend #401402) or anti-CAR (clone RmcB) antibody (Millipore #05-644) with (Kupffer cell staining) or without (hepatocyte staining) PE-conjugated anti-F4/80 (BioLegend #123109) for 30 min on ice. After rinsing with FACS buffer twice, the cells were treated with Alexa Fluor 488-conjugated anti-mouse IgG (Thermo Fisher Scientific #A21202) for 30 min on ice. After rinsing with Wash buffer twice, the cells were resuspended in FACS buffer and analyzed. To determine the pDC population, mice were sacrificed and perfused with DPBS. Spleens and livers were isolated, cut into small pieces and incubated in Collagenase D solution [2% FBS, HG 1:100, 5 M MgCl2/CaCl2 1:500, P/S/G 1:100, Collagenase D (Roche, #11088860001) 1 mg/ml final, DNase 1:500 (20 μg/ml, Roche) in RPMI] by shaking at 180 rpm at 37 °C for 30 min. Then, the tissues were mechanically disrupted. Splenic red blood cells were lysed with 0.83% NH4Cl. Hepatic immune cells for pDC analysis were isolated using lympholite-M (Cedarlane, #CL5031) according to the manufacturer's instruction. After rinsing with PBS several times, followed by Fc-blocking with anti-CD16/32 (BD Biosciences #553142), immune cells were immunophenotyped with PerCP-Cy5.5-conjugated anti-CD11c (eBioscience #45-0114-82), PE-conjugated anti-PDCA-1 (BioLegend #127009), and APC-conjugated anti-Siglec-H (BioLegend #129611). Samples were acquired on a BD Biosciences LSR-II and analyzed using FlowJo (Treestar). Hepatocytes were identified based on DNA content. Kupffer cells were defined as hepatic F4/80+ cells. pDCs in these studies were defined as CD11c+, PDCA-1+, and Siglec-H+ cells. Gating strategies for all the flow cytometry analyses are included in Supplementary Fig. 5.

**Quantification and statistical analysis**. All data were analyzed using Prism software (GraphPad Prism 7/8). The two-tailed Mann–Whitney test was used to analyze differences in viral burden. Kaplan–Meier survival curves were analyzed by the log rank test. The two-tailed student's t-test was used to analyze differences in CVB3 RNA replication, body weight loss, nuclei area quantitation, serum IFN-α levels, and serum ALT activity levels. $P$ values less than 0.05 were considered significant and are indicated in figures as follows: * $0.05 > p > 0.01$; ** $0.01 \geq p > 0.001$; *** $0.001 \geq p > 0.0001$; **** $p \leq 0.0001$.

**Reporting summary**. Further information on research design is available in the Nature Research Reporting Summary linked to this article.

## Data availability
Relevant data and/or materials not present are available upon reasonable request from T.K. (tkimura@scripps.edu) and/or J.L.W. (lwhitton@scripps.edu). Source data for the main figures is present in Supplementary Data 1.

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

## Acknowledgements
This work was supported by NIH grant R01AI114615 (JLW) and by American Heart Association Postdoctoral Fellowship 18POST33960190 to TK. This is manuscript number 29904 from the Scripps Research Institute.

## Author contributions
T.K.: design of experiments, execution of experiments, data preparation, data analysis, wrote paper, provided financial support. C.F.: execution of experiments, data preparation, data analysis, reviewed and commented on paper. J.L.W.: overview of design, assist with data analysis, wrote paper, provided financial support.

## Competing interests
The authors declare no competing interests
