## [Peer Review File · Communications Biology]

Reviewers' comments:

Reviewer #1 (Remarks to the Author):

This manuscript by Taishi Kimura et al, entitled "Hepatocytes trap and silence coxsackieviruses, protecting against systemic disease" sought to determine the role of hepatocytes in modulating coxsackievirus infection. It is well-established that the liver plays a significant role in clearing microbes from the host, however, a rigorous investigation of the role of hepatocytes in the abrogation of coxsackieviruses infection remains to be performed. This work demonstrates for the first time that hepatocytes play a major role in abrogating systemic coxsackievirus B3 infection. Using an albumin promoter(hepatocyte promoter)-specific deletion of the coxsackievirus-adenovirus receptor, the authors demonstrated that hepatocyte signaling and host defense function are critical for controlling systemic dissemination of coxsackievirus B3. Using imaging technologies, the authors demonstrate that hepatocytes trap and silence the virus.

This manuscript is novel and addresses an important question on the role of hepatocytes in host defense against viruses. I recommend this manuscript be accepted upon addressing a minor concern. Specifically, did the author examined the kupffer cells in the liver of their CARHEP KO mice. That data is not provided in the manuscript. Although the CAR KO was designed to be hepatocyte-specific, functional defects could occur in off-target cells (i.e. kupffer cells). The authors did show heart tissue as a control, demonstrating liver specificity of their KO, however, a Kupffer cells versus hepatocytes analysis is important to demonstrate hepatocyte-specific defects. Kupffer cells are critical in liver-associated host defense, thus to rule out a role for kupffer cells in this phenotype, the authors should show that those cells remain intact or at the very least, explain why they presumed they remained intact (preferably with supporting published data).

Moses T. Bility, PhD

Reviewer #2 (Remarks to the Author):

In this interesting and well-written paper, the authors show that mice with hepatocyte-specific knockout of the CVB3 receptor CAAR are at risk for increased morbidity and mortality with higher viremia and more severe systemic disease, including myocarditis. They interpret their findings as indicative of the ability of hepatocytes to 'absorb and silence' virus without supporting productive replication, and that although this results in mild liver disease with occasional hepatocyte apoptosis, that it is overall protective for the host and an unrecognized but potentially important aspect of innate immunity. The experiments are logical and well described, and the data are clearly presented and convincing in terms of the protection afforded the host by CAR expression in hepatocytes. However, there are concerns that the authors' may have underestimated the extent to which virus replicates in hepatocytes in the WT mice, the role of IRF1 in the systemic protection afforded by hepatocyte infection, as well as the exclusive use of the IP route for virus challenge. There are also concerns related to the broad nature of the speculation that this hepatocyte "sponge" effect extends to many other viral pathogens that are not strictly hepatotropic.

Specific comments:

1. The authors claim there is no productive CVB3 replication in hepatocytes (line 139), based on the absence of a difference in liver titer in WT and CAR-HEP-KO mice (lines 51-55). However, the liver titers increase >100-fold from days 1-4, and on day 4 there is 100-fold more virus in liver than in serum (Figs 1F and 2D). What explains this if its not replication in the liver? Is the virus replicating in another cell type (sinusoidal endothelium or Kupffer cells?) and is this masking the contribution of replication in hepatocytes which while dominant are not the only cell type in the liver?

2. The authors suggest that the inability of hepatocytes to "absorb" CVB3 resulted in an increase in serum titers, resulting in worsening systemic disease. If this is the case, why is there no difference in the amount of CVB3 in the livers of WT and CAR-HEP-KO mice as shown Fig 1? They conclude that hepatic absorption of CVB3 is enhanced in WT, but nowhere are there data showing a difference in the absorption of CVB3 between WT and CAR-KO mice.

3. The IRF1-KO experiment shows that virus can replicate in hepatocytes. How much does the IRF1 response to this in hepatocytes contribute to systemic IFN levels? Could it be that early, IRF1-induced IFN produced in hepatocytes protects systemically? The scale in Fig 1G doesn't provide insight into whether there are differences in serum IFN at day 1, when the IRF1 response in hepatocytes might be most important (authors' reference 20). Is the liver simply a sponge, or does it contribute to an active innate immune response that protects systemically?

4. Fas-mediated hepatocellular apoptosis induces hepatic inflammation (PMID: 11602613) and it is therefore surprising not to see some inflammatory cells in the livers of CV3B-infected mice with hepatocyte apoptosis. This cannot be evaluated well in the EDF 3 image of TUNL staining. Staining for cleaved caspase 3 by immunohistochemistry would allow a determination of whether inflammatory cells are recruited to sites of apoptotic cells. A wider view of liver histology than that shown in Fig. 2, to include multiple liver lobules, would also be helpful given the 3-4x ALT elevation.

5. All virus challenges were done via an IP route, in which virus is likely to traffic directly to the liver. Do hepatocytes play a similar role in IV or respiratory tract challenge?

6. Line 139-142: The text states "hepatocyte-specific CAR ablation ... (iv) protects against systemic disease in distant organs (myocarditis)". Is this really what the authors meant to say? The authors' data suggest hepatocytes protect, but that ablation enhances, rather than protects, against systemic disease (line 85).

Reviewer #3 (Remarks to the Author):

Kimura et. al in their manuscript titled "Hepatocytes trap and silence coxsackieviruses, protecting against systemic disease" employ a CAR (coxsackievirus-adenovirus receptor) KO mice model system to explore the functions of hepatocytes in removing coxsackievirus B3 (CVB3). They demonstrate that CAR KO mice are unable to capture circulating CVB3, which leads to a dramatic early rise of virus in the blood (viremia), followed by accelerated systemic disease and increased mortality in comparison to the genetically-intact mice, where the trapping and rapid silencing of virus leads to widespread alterations in hepatocyte morphology. They conclude that hepatocytes can act as a "sponge", trapping and suppressing viruses without triggering overt hepatitis, and suggest that hepatocytes, as an integral component of the innate immunity, may internalize and silence a large variety of different viruses, diminishing pathogen burden and protecting against disease.

I think this study is well-designed and the results will be interesting to people, especially for those who investigate on the hepatocyte or hepatocyte-related areas including the innate immunity by hepatocytes.

There are some questions need to be addressed

1. Line 160-173 The author mentioned that transient elevation of serum transaminase by CVB3 were also observed in other virus infection, thus suggesting that this previously-unidentified hepatocyte activity should be considered an integral part of the innate response to viral infection. I

think some suggestion by the authors is short of evidence, especially the investigations performed by other labs. It is too early to make such claim since many agents (factors) can trigger the transaminase increase in the serum.

2. Line 4-5, "a function that is currently attributed to cells that reside in the hepatic sinusoids" would be much better as "a function that is currently attributed to cells residing in the hepatic sinusoids" to avoid double attributive clauses in a sentence.

Reviewer #1 wrote that “*This manuscript is novel and addresses an important question on the role of hepatocytes in host defense against viruses. I recommend this manuscript be accepted upon addressing a minor concern. Specifically, did the author examined the kupffer cells in the liver of their CARHEP KO mice. That data is not provided in the manuscript. Although the CAR KO was designed to be hepatocyte-specific, functional defects could occur in off-target cells (i.e. kupffer cells). The authors did show heart tissue as a control, demonstrating liver specificity of their KO, however, a Kupffer cells versus hepatocytes analysis is important to demonstrate hepatocyte-specific defects. Kupffer cells are critical in liver-associated host defense, thus to rule out a role for kupffer cells in this phenotype, the authors should show that those cells remain intact or at the very least, explain why they presumed they remained intact (preferably with supporting published data).*”

RESPONSE: The Alb-Cre mice have been used in many published studies, and it is widely accepted that the Cre activity is limited to hepatocytes. However, we agree that the reviewer’s question is an important one, and ***in response we have made three changes to the manuscript.*** **First**, on page 2 of the revised manuscript, we have cited (and briefly discussed) a published paper (Takehara *et al.*, 2004) in which Alb-Cre activity was shown to occur in hepatocytes, but had no detectable effect on non-hepatocyte cells in the liver. **Second**, a new experimental dataset has been included. We have carried out flow cytometry analyses, which show that, in CAR^{HEP}KO mice, cell-surface CAR expression is ablated from hepatocytes (Figure 1d) but is maintained in Kupffer cells (Figure 1e). **Third**, another new experimental dataset – CAR immunohistochemistry – has been added to the revised paper (see three new panels, included as Figure 1f). CAR protein expression is abundant in CAR WT mice, but is dramatically reduced in CAR^{HEP}KO livers, in which the residual CAR expression is sinusoidal.

Please note that, as a result of the inclusion of these new data, we decided to divide the original figure 1 into two figures (figures 1 & 2). Consequently, the revised manuscript has five figures, rather than 4.

Reviewer #2 found the paper “*interesting and well-written*”, and opined that “*the data are clearly presented and convincing in terms of the protection afforded the host by CAR expression in hepatocytes*”. However, he/she had 6 concerns/comments:

1. *The authors claim there is no productive CVB3 replication in hepatocytes (line 139), based on the absence of a difference in liver titer in WT and CAR-HEP-KO mice (lines 51-55). However, the liver titers increase >100-fold from days 1-4, and on day 4 there is 100-fold more virus in liver than in serum (Figs 1F and 2D). What explains this if its not replication in the liver? Is the virus replicating in another cell type (sinusoidal endothelium or Kupffer cells?) and is this masking the contribution of replication in hepatocytes which while dominant are not the only cell type in the liver?*

RESPONSE: With regard to hepatic virus titers, we have two responses. **First**, we did not suggest that CVB3 does not replicate in the liver. We argued that it does not (to any significant extent) replicate in hepatocytes. We agree with the reviewer’s suggestion that the virus replicates in cells in the sinusoids, and we had indicated this in the original version of the paper, where we wrote that the viral VP1 protein: “*was observed in the sinusoids (Figure 1h and Supplemental movie 1), suggesting that, in genetically-intact animals, intrahepatic CVB3 replication may be limited to sinusoidal cells*”. The legend to Fig 1h (now Fig 2g) read (and still reads) “*VP1 (green) was not found in WT hepatocytes, but signal was observed within sinusoids (blue = nuclei; red = F-actin).*” **Second**, our claim that CVB3 does not replicate in hepatocytes is not based solely on the absence of a difference in liver titer in WT and CAR-HEP-KO mice; it is also based on data from tissue culture studies that are presented in Fig 2.

Nevertheless, to make our point regarding sinusoidal (non-hepatocyte) replication of CVB3 even more clearly, ***we have made the following six changes to the manuscript.***

- The new CAR flow cytometry (Fig 1d & e) shows that, in the CAR^{HEP}KO mice, CAR expression is ablated from hepatocytes, but is retained on Kupffer cells.
- The new CAR immunohistochemistry (Fig 1f) demonstrates that, in CAR^{HEP}KO mice, CAR signal is observed only in sinusoids, consistent with viral replication (which requires CAR-mediated uptake) occurring in sinusoids.
- We have added the following two sentences to page 3 of the text, after presenting Fig 2 panels d & e: *“In both strains, hepatic virus titers rose very substantially between days 1 and 4 p.i., indicating that CVB3 does replicate in the liver, albeit not in hepatocytes. The site(s) of hepatic CVB3 replication will be addressed below”*.
- A few sentences later, we have slightly modified an existing sentence to read (added text underlined): *“We also used confocal microscopy of vibratome sections to identify the viral VP1 protein in the livers of WT and IRF1KO mice at d1 p.i. VP1 signal was very rarely detectable within hepatocytes of WT mice, and instead was observed in the sinusoids (Figure 2g and Supplemental movie 1), suggesting that, in genetically-intact animals, intrahepatic CVB3 replication may occur in, and be limited to, sinusoidal cells”*.
- We have added new data (new Extended data Figure 2) showing that, at d2 p.i. of WT mice and CAR^{HEP}KO mice, viral signal (both VP1, and ds RNA) is observed in sinusoids, rather than in hepatocytes.
- Related to the above new Extended Data figure, we have added, to the main text (page 3) the sentence: *“Our suggestion that CVB3 replicates in hepatic sinusoids, rather than in hepatocytes, is supported by analyses of hepatic CVB3 distribution at d2 p.i.. As shown in Extended Data Fig. 2, in both WT and CAR^{HEP}KO mice, VP1 was present mainly in sinusoids, and the same was true for dsRNA, identified by the use of J2 antibody”*.

Finally, the reviewer compares the virus titers in the livers to those of sera. However, please note that hepatic virus titers are shown as PFU/gram, while serum virus titers are shown as PFU/mL. Therefore, these two sets of virus titer data are not directly comparable.

2. *The authors suggest that the inability of hepatocytes to "absorb" CVB3 resulted in an increase in serum titers, resulting in worsening systemic disease. If this is the case, why is there no difference in the amount of CVB3 in the livers of WT and CAR-HEP-KO mice as shown Fig 1? They conclude that hepatic absorption of CVB3 is enhanced in WT, but nowhere are there data showing a difference in the absorption of CVB3 between WT and CAR-KO mice.*

RESPONSE: We see no difference in liver titers between the two strains because, regardless of whether the virus is absorbed by hepatocytes (WT mice) or is not absorbed by hepatocytes (CAR^{HEP}KO mice), no viral replication will occur in hepatocytes. Moreover, as shown in the new Extended Data Fig 2, the viral signals (VP1 and ds RNA) observed in sinusoids are similar in quantity in both mouse strains, suggesting that infection of sinusoidal cells in the WT mice is already at saturation level. Thus, we suggest that excess “free” virus may be present in the sinusoids of the CAR^{HEP}KO mice, but all susceptible CAR-expressing sinusoidal cells have already been infected.

3. *The IRF1-KO experiment shows that virus can replicate in hepatocytes. How much does the IRF1 response to this in hepatocytes contribute to systemic IFN levels? Could it be that early, IRF1-induced IFN produced in hepatocytes protects systemically? The scale in Fig 1G doesn't provide insight into whether there are differences in serum IFN at day 1, when the IRF1 response in hepatocytes might be most important (authors' reference 20). Is the liver simply a sponge, or does it contribute to an active innate immune response that protects systemically?*

RESPONSE: Fig 1G did not contain any IFN-related data. I think the reviewer is referring to Fig 2g (now, Fig 3g). In summary, at early time points (*d1-2 p.i.*), viremia is higher in the CAR^{HEP}KO mice

than in WT mice. We attribute this to the inability of CAR-deficient hepatocytes to absorb the virus. To the best of our understanding, the reviewer is asking if, instead, the difference in viremia might be attributed to an early burst of systemic T1IFN that results from activation of hepatocyte IRF-1; perhaps (he/she suggests) the hepatic uptake of CVB3 by WT hepatocytes leads to IRF-1 stimulation (and elevated systemic T1IFNs) that does not take place in the hepatocytes of the CAR^{HEP}KO mice. We consider this unlikely, because – as shown in Fig 3g – the serum IFN- α 2/ α 4 level in the latter mice was markedly higher than in WT animals. With regard to the reviewer's concern about the scale in Fig 3g potentially obscuring an effect at d1 post-infection: we feel that the data show clearly that all serum samples at the d1 time point had near-undetectable levels of T1IFNs. However, to more directly address the reviewer's concern, please note that the near-undetectable levels of T1IFNs at early time points in genetically-intact mice were (and are still) displayed, at a "magnified" scale, in Fig. 5c.

4. *Fas-mediated hepatocellular apoptosis induces hepatic inflammation (PMID: 11602613) and it is therefore surprising not to see some inflammatory cells in the livers of CV3B-infected mice with hepatocyte apoptosis. This cannot be evaluated well in the EDF 3 image of TUNL staining. Staining for cleaved caspase 3 by immunohistochemistry would allow a determination of whether inflammatory cells are recruited to sites of apoptotic cells. A wider view of liver histology than that shown in Fig. 2, to include multiple liver lobules, would also be helpful given the 3-4x ALT elevation.*

RESPONSE: The reviewer's surprise at the absence of overt hepatitis is understandable, especially since (i) the oil red O staining shows massive involvement of hepatocytes and (ii) there was a transient rise in ALT, indicative of hepatocyte damage. However, we feel that it is important to stress that our findings regarding the absence of hepatitis during CVB3 infection mirror those published by others. We had already included some related references (numbered 15-19), but we have added a further citation (Wang *et al.*, 2010, cited as new ref 23). Those authors investigated the outcome of CVB3 infection in WT mice and in mice lacking the T1IFN receptor. They observed no pathology by H&E staining of the WT livers, while in the T1IFNRKO animals the authors reported "severe hepatic cell necrosis without inflammatory cell infiltration". Thus, **we have slightly expanded a clause on page 5**, changing it from "confirming that CVB3 does not trigger florid hepatitis¹⁹", to "this observation confirms others' findings that, in genetically-intact animals, CVB3 does not trigger florid hepatitis^{19,23}". **In addition, we have carried out the cleaved caspase 3 IHC experiment proposed by the reviewer, and the data are included in Extended Data Fig 3 (previously, ED Fig 4). Related to this, we have altered the main text** (see page 5 of the revised submission). The original sentence was: "Given the absence of inflammation, we also performed TUNEL assays to assess hepatic apoptosis (Extended Data Fig. 4). Apoptotic cells were detected in liver sections from WT mice throughout infection, but were very rarely observed in CARHEPKO livers". This has been replaced with: "We also performed two assays to assess hepatic apoptosis: TUNEL, and immunohistochemistry to detect cleaved caspase 3 (Extended Data Fig. 3). The TUNEL assay showed that apoptotic cells were present, albeit at low numbers, in liver sections from CVB3-infected WT mice, but they were almost undetectable in infected CARHEPKO livers. The cleaved caspase 3 data confirm the TUNEL findings; CVB3 rarely triggers apoptosis in WT livers. Moreover, the data support and extend the conclusion drawn from Figure 4e; at 4 days p.i., when ALT levels are highest, hepatic inflammation is minimal, and this is true even adjacent to the infrequent apoptotic foci".

Finally, the reviewer suggested that "A wider view of liver histology than that shown in Fig. 2, to include multiple liver lobules, would also be helpful...". Fig 2 did not contain any hepatic histology; we believe that the reviewer was referring to original Fig 3 (now Fig 4). In that regard, we note that Fig 4a shows that – at least macroscopically – all visible lobules of the liver appear to be abnormal (in WT mice). We believe that this, together with the addition of the new citation (Wang *et al.*) showing that the lack of hepatic inflammation during CVB3 infection is unsurprising, render it unnecessary to individually evaluate each of the liver's lobules.

5. All virus challenges were done via an IP route, in which virus is likely to traffic directly to the liver. Do hepatocytes play a similar role in IV or respiratory tract challenge?

RESPONSE: We have not carried out IV or respiratory challenges. Therefore, **we have added a qualification to the text**, writing, on page 7 (underlined text has been added to the revision): “Thus, following i.p. infection, hepatocytes act as a shield against CVB3, but they also may do so during natural infections by many other “non-hepatitis” viruses”.

6. Line 139-142: The text states “hepatocyte-specific CAR ablation ... (iv) protects against systemic disease in distant organs (myocarditis)”. Is this really what the authors meant to say? The authors’ data suggest hepatocytes protect, but that ablation enhances, rather than protects, against systemic disease (line 85).

RESPONSE: We thank the reviewer for picking up on this error, and we apologize for its presence! **We have changed the text** to read (page 6) “hepatocyte-specific CAR ablation.... (iv) enhances systemic disease in distant organs (myocarditis)...”

Reviewer #3: wrote that “this study is well-designed and the results will be interesting to people”, and had two questions.

1. Line 160-173 The author mentioned that transient elevation of serum transaminase by CVB3 were also observed in other virus infection, thus suggesting that this previously-unidentified hepatocyte activity should be considered an integral part of the innate response to viral infection. I think some suggestion by the authors is short of evidence, especially the investigations performed by other labs. It is too early to make such claim since many agents (factors) can trigger the transaminase increase in the serum.

RESPONSE: We feel that our data strongly suggest that, during CVB3 infection, hepatocytes can be viewed as innate effector cells which act by absorbing and silencing the virus. Given that (as cited in the manuscript) (i) transient elevations of transaminases have been observed in many other infections by non-hepatotropic viruses and (ii) others have shown that hepatocytes in tissue culture can absorb and silence several viruses other than CVB3, it is reasonable to propose that the phenomenon may be generalizable. We do not claim to have proved it, but we feel that speculation is both valid and interesting, especially as it provides an intriguing evolutionary reason for the astonishing regenerative capacity of hepatocytes. However, **we have slightly modified the sentence** (see page 7 of revised manuscript), changing “hepatocyte activity should be considered...” to “hepatocyte activity might be considered...”

2. Line 4-5, “a function that is currently attributed to cells that reside in the hepatic sinusoids” would be much better as “a function that is currently attributed to cells residing in the hepatic sinusoids” to avoid double attributive clauses in a sentence.

RESPONSE: **We have made the requested change** (see page 1 of revised manuscript).

Professor

REVIEWERS' COMMENTS:

Reviewer #1 (Remarks to the Author):

The authors addressed my concerns; I recommend acceptance of the manuscript.

Reviewer #2 (Remarks to the Author):

The authors have fully addressed the issues raised in the initial review of this interesting manuscript.

Reviewer #3 (Remarks to the Author):

After going over the paper, I am satisfied that the authors have addressed my questions and I have no further concern.